# Molecular Pathogenesis of Pancreatic Ductal Adenocarcinoma: Impact of *miR-30c-5p* and *miR-30c-2-3p* Regulation on Oncogenic Genes

**DOI:** 10.3390/cancers12102731

**Published:** 2020-09-23

**Authors:** Takako Tanaka, Reona Okada, Yuto Hozaka, Masumi Wada, Shogo Moriya, Souichi Satake, Tetsuya Idichi, Hiroshi Kurahara, Takao Ohtsuka, Naohiko Seki

**Affiliations:** 1Department of Digestive Surgery, Breast and Thyroid Surgery, Graduate School of Medical and Dental Sciences, Kagoshima University, Kagoshima 890-8520, Japan; takako-t@m.kufm.kagoshima-u.ac.jp (T.T.); k6958371@kadai.jp (Y.H.); k8911571@kadai.jp (M.W.); k8745685@kadai.jp (S.S.); k3352693@kadai.jp (T.I.); h-krhr@m3.kufm.kagoshima-u.ac.jp (H.K.); takao-o@kufm.kagoshima-u.ac.jp (T.O.); 2Department of Functional Genomics, Chiba University Graduate School of Medicine, Chiba 260-8670, Japan; reonaokada@chiba-u.jp; 3Department of Biochemistry and Genetics, Chiba University Graduate School of Medicine, Chiba 260-8670, Japan; moriya.shogo@chiba-u.jp

**Keywords:** pancreatic ductal adenocarcinoma, microRNA, tumor-suppressor, *miR-30c-5p*, *miR-30c-2-3p*, *TOP2A*

## Abstract

**Simple Summary:**

A total of 10 genes (*YWHAZ, F3, TMOD3, NFE2L3, ENDOD1, ITGA3, RRAS, PRSS23, TOP2A*, and *LRRFIP1*) were identified as tumor suppressive *miR-30c-5p* and *miR-30c-2-3p* targets in pancreatic ductal adenocarcinoma (PDAC), and expression of these genes were independent prognostic factors for patient survival. Furthermore, aberrant expression of *TOP2A* and its transcriptional activators (*SP1* and *HMGB2*) enhanced malignant transformation of PDAC cells.

**Abstract:**

Pancreatic ductal adenocarcinoma (PDAC) is one of the most aggressive types of cancer, and its prognosis is abysmal; only 25% of patients survive one year, and 5% live for five years. MicroRNA (miRNA) signature analysis of PDAC revealed that both strands of pre-*miR-30c* (*miR-30c-5p*, guide strand; *miR-30c-2-3p*, passenger strand) were significantly downregulated, suggesting they function as tumor-suppressors in PDAC cells. Ectopic expression assays demonstrated that these miRNAs attenuated the aggressiveness of PDAC cells, e.g., cell proliferation, migration, and invasiveness. Through a combination of in silico analyses and gene expression data, we identified 216 genes as putative oncogenic targets of *miR-30c-5p* and *miR-30c-2-3p* regulation in PDAC cells. Among these, the expression of 18 genes significantly predicted the 5-year survival rates of PDAC patients (*p* < 0.01). Importantly, the expression levels of 10 genes (*YWHAZ, F3, TMOD3, NFE2L3, ENDOD1, ITGA3, RRAS, PRSS23, TOP2A*, and *LRRFIP1*) were found to be independent prognostic factors for patient survival (*p* < 0.01). We focused on *TOP2A* (DNA Topoisomerase II Alpha) and investigated its potential as a therapeutic target for PDAC. The overexpression of *TOP2A* and its transcriptional activators (*SP1* and *HMGB2*) was detected in PDAC clinical specimens. Moreover, the knockdown of *TOP2A* enhanced the sensitivity of PDAC cells to anticancer drugs. Our analyses of the PDAC miRNA signature and tumor-suppressive miRNAs provide important insights into the molecular pathogenesis of PDAC.

## 1. Introduction

Pancreatic ductal adenocarcinoma (PDAC), which accounts for the majority of pancreatic cancer (approximately 90%), is one of the most malignant tumors in the world and the third leading cause of cancer death, characterized by frequent metastases and profound chemotherapy resistance [1,2]. The 5-year relative survival rate for pancreatic cancer is the lowest among cancers—only 9% across all stages, and the death rates for pancreatic cancers have risen over the past decade [3]. The current standard treatments for pancreatic cancers continue to rely on surgical resection and cytotoxic therapies; however, only less than 20% of patients are eligible to complete surgical resection, and many will relapse and die within one year [3,4,5,6]. To make matters worse, owing to the asymptomatic nature of the early stage of the disease and the absence of efficient diagnostic methods, most patients with PDAC present with the locally advanced and inoperable disease at diagnosis [3,4,5,6]. Thus, it is essential to search for new diagnostic markers and develop new therapeutic strategies based on the latest genomic analyses of PDAC.

Sequencing through the human genome project has revealed that a vast number of non-coding RNAs are transcribed from the human genome and act as pivotal players in various cellular pathways in physiological and pathological conditions [7]. MicroRNAs (miRNAs) are members of endogenous non-coding RNAs. They are single-stranded RNA, 19 to 23 nucleotides in length, and their functions include fine-tuning of RNA expressions by degradation or translational inhibition of the target RNA transcripts [8,9]. A unique feature of miRNAs is that a single miRNA can control a vast number of RNA transcripts (protein-coding genes and non-coding genes), and, conversely, one mRNA can be the target of numerous miRNAs in diseased and normal cells [8,9]. Bioinformatic studies have shown that more than half of the genes expressed in cells are under the control of miRNAs [10,11]. Given this nearly ubiquitous expression, aberrantly expressed miRNAs can cause perturbations of RNA networks within cells, and these factors are among those that can transform normal cells into cancer cells. Consistently, many studies have shown that aberrant expressions of miRNAs are closely involved in oncogenesis [12,13,14].

Over the last decade, by applying the latest genomics analysis methods, the search for miRNAs whose expressions are altered in PDAC cells has been energetically carried out [15,16,17]. A vast number of studies in PDAC cells have revealed that dysregulated miRNAs are deeply involved in the malignant transformation of cancer cells, e.g., cell cycle runaway, suppression of apoptosis, promotion of cell invasion and metastasis, and acquisition of drug resistance [15,16,17].

Our recent studies have shown that some miRNA passenger strands derived from the pre-miRNA-duplex act as tumor-suppressive miRNAs and their target genes are closely involved in oncogenesis [18,19,20,21,22]. The involvement of miRNA passenger strands in oncogenesis is a new concept in cancer research. Thus, the analysis of both miRNA strands is essential in searching for the molecular pathogenesis of human cancers [23]. Using RNA-sequencing technology, we constructed the miRNA expression signature of PDAC [24]. Analysis of our PDAC signature revealed that both miRNA strands derived from pre-*miR-30c* (*miR-30c-5p*, the guide strand; *miR-30c-2-3p*, the passenger strand) were significantly downregulated in PDAC tissues.

The miR-30-family is located on three chromosomal regions (chromosome 1: miR-30c-1 and miR-30e; chromosome 6: miR-30c-2 and miR-30a; chromosome 8: miR-30b and miR-30d) and consists of 12 different mature miRNAs [25]. The seed sequences of the guide strands of the miR-30 family are identical (*GUAAACA*). In contrast, there are variations in the seed sequences of the passenger strands of the miR-30 family. However, the mature sequences of miR-30c-1-3p (*CUGGGAGAGGGUUGUUUACUCC*) and *miR-30c-2-3p* (*CUGGGAGAAGGCUGUUUACUCU*) are almost identical. 

Accumulating evidence suggests that guide strands of the miR-30 family act as critical players (tumor-suppressor or oncogenic) in a wide range of human cancers [26,27,28,29]. In contrast, there have been almost no reports on the role of miR-30 family passenger strands in PDAC research. In this study, we focused on both strands of the pre-*miR-30c* (*miR-30c-5p*, the guide strand; *miR-30c-2-3p*, the passenger strand) and sought to identify their target genes, which are intimately involved in the molecular pathogenesis of PDAC. A total of 10 genes (*YWHAZ*, *F3*, *TMOD3*, *NFE2L3*, *ENDOD1*, *ITGA3*, *RRAS*, *PRSS23*, *TOP2A*, and *LRRFIP1*) were identified as being significantly predictive of worse prognosis (5-year survival rate) in patients with PDAC. Furthermore, we discussed the functional significance of *TOP2A* in PDAC cells.

## 2. Results

### 2.1. Downregulation of miR-30c-5p and miR-30c-2-3p in PDAC Clinical Specimens and Cell Lines

To confirm the miRNA expression signature of PDAC, we evaluated the expression levels of *miR-30c-5p* and *miR-30c-2-3p* in PDAC clinical specimens. The clinical features of the patients are summarized in Appendix A.

The expression levels of *miR-30c-5p* and *miR-30c-2-3p* were significantly downregulated in PDAC tumor tissues compared with matched normal tissues (*p* = 0.0045 and *p* = 0.0013, respectively; Figure 1A). Their expression was also confirmed to be low in two PDAC cell lines: PANC-1 and SW1990 (Figure 1A). Furthermore, there was a positive correlation between *miR-30c-5p* and *miR-30c-2-3p* expression among the clinical samples (Spearman’s rank analysis; *r* = 0.832, *p* < 0.001; Figure 1B).

### 2.2. Effects of Ectopic Expression of miR-30c-5p and miR-30c-2-3p on Cell Proliferation, Migration, and Invasion in PDAC Cells

The antitumor activities of *miR-30c-5p* and miR-30c-3p were assessed by transfecting these miRNAs into PANC-1 and SW1990 cells. Specifically, the suppression of cell proliferation was observed following miR-30c-3p transfection (Figure 2A). Furthermore, ectopic expression of *miR-30c-5p* and miR-30c-3p suppressed cell migration and cell invasion (Figure 2B,C).

### 2.3. Identification of miR-30c-5p and miR-30c-2-3p Targeted Genes in PDAC Cells

To screen for genes regulated by *miR-30c-5p* and *miR-30c-2-3p*, we obtained RNA microarray data for PANC-1 and SW1990 cells transfected with *miR-30c-5p* and *miR-30c-2-3p*. These data were combined with gene expression datasets from Gene Expression Omnibus (GEO; GSE15471) and data obtained from the TargetScanHuman database (release 7.2), which provides data on the putative targets of each miRNA. A total of 216 genes were identified as putative oncogenic targets of *miR-30c-5p* and *miR-30c-2-3p* (51 genes for *miR-30c-5p* and 165 genes for miR-30c-3p; Appendix A). Among these 216 genes, the expressions of 18 genes showed statistically significant correlations with the 5-year overall survival rates of patients with PDAC (*p* < 0.01; Table 1 and Figure 3). The expression levels of these target genes were reevaluated using the GEO dataset, and all the genes were confirmed to be upregulated in PDAC tissues compared to normal pancreatic tissues (Figure 4). We also validated the gene expression levels of these 18 genes using another expression dataset (GSE28735). The analysis data is shown in Appendix A. Two datasets revealed that the expressions of these genes were upregulated in cancerous tissues. 

The expression levels of these proteins were verified by “The Human Protein Atlas” database (https://www.proteinatlas.org/). The results of immunostaining for these proteins are summarized in Appendix A. 

Furthermore, in order to predict the functions of these genes, gene ontology (GO) classification was performed using the GeneCodis database. According to the GO classification, it contained significant genes that were classified as protein binding (GO:0005515) and DNA binding (GO:0003677). The analysis data is shown in Appendix A.

Furthermore, multivariate analysis identified 10 genes (*YWHAZ*, *F3*, *TMOD3*, *NFE2L3*, *ENDOD1*, *ITGA3*, *RRAS*, *PRSS23*, *TOP2A*, and *LRRFIP1*) as independent prognostic factors for patient survival (*p* < 0.01; Figure 5). For subsequent analyses, we focused on *TOP2A* as an important oncogene based on previous reports [30]. *TOP2A* is an enzyme that can control topological state and DNA replication by breaking and recombination of DNA strands. It has been reported that aberrant expression of *TOP2A* is implicated in malignant transformation of cancer cells, e.g., proliferation, metastasis, and chemotherapeutic drug resistance [30]. Interestingly, a recent study showed that *TOP2A* interacted directly with β-catenin, activating the β-catenin pathways. These events promoted Epithelial-Mesenchymal Transition (EMT) pathways and aroused metastasis of PDAC cells [31].

### 2.4. Direct Regulation of TOP2A by miR-30c-2-3p in PDAC Cells

In PDAC cells transfected with *miR-30c-2-3p*, both *TOP2A* mRNA expression and *TOP2A* protein expression were significantly suppressed (Figure 6A,B). 

In addition, to validate whether *miR-30c-2-3p* directly binds to *TOP2A* in PDAC cells, we performed dual-luciferase reporter assays. We designed plasmid vectors encoding the partial sequence of the 3′-UTR of *TOP2A*. One of them included the predicted *miR-30c-2-3p* target site (“wild-type”), and the other lacked the target site (“deletion-type”; Figure 6C). The luciferase activity was significantly decreased following co-transfection of *miR-30c-2-3p* and a vector carrying the wild-type 3′-UTR of *TOP2A*. In contrast, no decrease in luminescence was observed following transfection of the deletion-type vector (Figure 6C). These findings indicate that *miR-30c-2-3p* directly binds to the 3′-UTR of *TOP2A*.

### 2.5. Incorporation of miR-30c-5p and miR-30c-2-3p into the RNA-Induced Silencing Complex (RISC) in PDAC Cells

Ago2 is an essential component of the RISC, which binds to miRNAs. Whether the transfected miRNAs were incorporated into RISC in PDAC cells was analyzed by immunoprecipitation using an Ago2 antibody. In *miR-30c-5p* transfected cells (PANC-1 and SW1990), it was confirmed that a large amount of *miR-30c-5p* was incorporated into RISC (Appendix A). Similarly, it was confirmed that *miR-30c-2-3p* transfected into PDAC cells was incorporated into RISC (Appendix A). From these facts, it was shown that the transfected miRNAs (*miR-30c-5p* and *miR-30c-2-3p*) were indeed functioning in the PDAC cells.

### 2.6. Effects of TOP2A Knockdown on Cell Proliferation, Migration, and Invasion in PDAC Cells

To assess the oncogenic functions of *TOP2A* in PDAC cells, we conducted knockdown assays with small interfering RNAs (siRNAs). The mRNA and protein expressions were effectively suppressed by si*TOP2A*-1 and si*TOP2A*-2 transfection into PANC-1 and SW1990 cells (Figure 7A).

Additional functional assays were performed with these siRNAs, and the results showed statistically significant suppression of cell proliferation by si*TOP2A* transfection in both cell lines (Figure 7B).

Moreover, cell migration and invasion activities were significantly downregulated by si*TOP2A* transfection in PDAC cells (Figure 7C,D).

We also investigated if the cisplatin sensitivity of pancreatic cancer cells could be increased by *TOP2A* knockdown. Co-treatment experiments showed an increased sensitivity to cisplatin in si*TOP2A*-treated cells (Figure 7E). Cisplatin sensitivity significantly increased from 5.259 μM (half maximal inhibitory concentration; IC_50_) to 3.22 μM (IC_50_).

### 2.7. Clinical Significance of Transcriptional Regulators of TOP2A in PDAC

Using the GSE15471 and TCGA-Pancreatic adenocarcinoma (PAAD) datasets, we analyzed the clinical importance of *TOP2A*-associated genes (*HDAC1, HDAC2, HMGB1, HMGB2, NFYA, NFYB, NFYC*, and *SP1*). These transcriptional regulators of *TOP2A* have been identified in a previous study [30].

In the GSE15471 dataset, the expression of the genes (except *HDAC2*) was significantly elevated in PDAC cancer tissues compared to normal tissues (*p* < 0.01; Figure 8). Furthermore, Kaplan–Meier survival analysis of the eight genes revealed that *HMGB2* expression and *SP1* expression were significantly predictive of the 5-year survival rate of patients with PDAC (*p* < 0.01; Figure 8).

### 2.8. Overexpression of TOP2A, HMGB2, and SP1 in PDAC Clinical Specimens

The expressions of *TOP2A*, *HMGB2*, and *SP1* protein were evaluated in PDAC clinical specimens by immunohistochemistry. The overexpression of the three proteins in the nuclei of cancer cells from PDAC clinical specimens was detected (Figure 9). The overexpression of these proteins (*TOP2A*, *HMGB2*, and *SP1*) was not observed in noncancerous tissues. Ki-67 was used as a marker for cell proliferating cells.

## 3. Discussion

Over the past decade, miRNAs aberrantly expressed in cancer cells have been identified using the latest genomic technologies: microarray, PCR-based array, and RNA-sequencing. As a result, the development of cancer diagnostics and treatment strategies based on miRNA has been accelerating [12,13,14]. The general concept of miRNA biogenesis involves the degradation of the miRNA passenger strands derived from the pre-miRNA-duplex [8,9]. Contrary to this theory, recent studies have shown that some miRNA passenger strands have tumor-suppressive functions, and their target genes can contribute to cancer progression, metastasis, and drug resistance in several types of cancers [18,19,20,21,22,23]. Our recent studies of PDAC cells revealed that *miR-216a-3p*, *miR-216b-3p, miR-148a-5p,* and *miR-130b-5p* were significantly downregulated in PDAC tissues using our RNA-sequencing based signature expression patterns [24,32,33]. Ectopic expression assays demonstrated that their expression attenuated the malignant phenotypes of PDAC cells and that the genes they regulate were aberrantly expressed in PDAC clinical specimens. Thus, they function as cancer-promoting genes, such as *FOXQ1, PHLDA2, LPCAT2, AP1S3*, and *EPS8* [24,32,33]. These studies suggest that miRNA analyses that include passenger strands will be crucial to understand the molecular pathogenesis of PDAC.

Based on our PDAC miRNA expression signature, we focused on both strands of pre-*miR-30c* (*miR-30c-5p* and *miR-30c-2-3p*). Our ectopic expression assays revealed that both miRNA strands attenuated cancer cell proliferation, migration, and invasion in PDAC cells. Previous studies have shown the downregulation of *miR-30c-5p* in several types of cancers, and the oncogenes it regulates have been implicated in various cancer pathways, such as cell proliferation, metastasis, and drug resistance [26,27]. For example, the overexpression of DLEU2 (lncRNA) functions as an adsorbed RNA that suppresses *miR-30c-5p* and subsequently activates the Akt signaling pathway in laryngeal squamous cell carcinomas [34]. In gastric cancer, the expression of *miR-30c-5p* has been shown to be significantly reduced in cancer tissues, and it also suppresses both metastasis and epithelial-to-mesenchymal transition through targeting *MTA1* [35].

Some reports have suggested that *miR-30c-2-3p* is involved in human cancers. In the human genome, miR-30c-2 and miR-30a are located close to the chromosomal 6q13 region. In clear cell renal cell carcinoma (ccRCC), *miR-30c-2-3p* and miR-30a-3p are downregulated in *von Hippel-Lindau* (VHL)-deficient cancer tissues, and both miRNAs specifically bind to hypoxia inducible factor-2α (HIF2A) and inhibit the expression of its transcripts [36]. Moreover, the constitutive expression of HIF2A enhances tumor growth in ccRCC [36]. In breast cancer, the expression levels of *miR-30c-2-3p* are characterized across patients with different molecular subtypes; lower expressions of *miR-30c-2-3p* are detected in human epidermal growth factor receptor 2 (HER2)-positive and triple-negative groups compared to estrogen receptor (ER)-positive groups [37]. It has been found that the expression of *miR-30c-2-3p* negatively regulates nuclear factor kappa B signaling and cell cycle progression in breast cancer cells via targeting TNFRSF1A-associated via death domain (TRADD) and cyclin E1 (CCNE1), respectively [37]. Recent studies have shown that non-coding RNAs, LncRNA-CCAT1, and has-circ-0072995 contribute to the aggressiveness of cancer cells by acting as competing endogenous RNAs for *miR-30c-2-3p* in hepatocellular carcinoma and breast cancer, respectively [38].

In the present study, we first revealed that both strands of pre-miR-30c-2 (*miR-30c-5p* and *miR-30c-2-3p*) acted as tumor-suppressive miRNAs in PDAC cells. Our next goal was to search for novel cancer-promoting genes in PDAC cells controlled by these tumor-suppressive miRNAs. Our miRNA target search strategy successfully identified 18 genes whose expression was significantly predictive of 5-year survival rates of patients with PDAC. Importantly, the expression levels of 10 genes (*YWHAZ*, *F3*, *TMOD3*, *NFE2L3*, *ENDOD1*, *ITGA3*, *RRAS*, *PRSS23*, *TOP2A*, and *LRRFIP1*) were found to act as independent prognostic factors for short patient survival times.

Among these genes, *YWHAZ* (14-3-3ζ) is a member of the 14-3-3 proteins and has multiple functions in various signal transduction pathways [39]. The overexpression of *YWHAZ* has been reported in a wide range of cancers, and its aberrant expression contributes to cancer progression and drug resistance [39]. In PDAC cells, tumor-associated macrophages invade cancer tissues after anticancer drug treatment, and the tumor-associated macrophage-secreted *YWHAZ* is shown to contribute to chemotherapy resistance [40]. Our recent studies identified the overexpression of *ITGA3* in PDAC and showed that its expression enhanced cancer cell migration and invasiveness through the activation of several oncogenic pathways [41]. Notably, *ITGA3* and ITGB1 expression was directly regulated by tumor-suppressive miR-124-3p in PDAC cells [41]. Further detailed analysis of the genes regulated by *miR-30c-2-3p* will contribute to our understanding of the molecular networks involved in PDAC.

In this study, we focused on *TOP2A* for further analysis in PDAC cells. The *TOP2A* enzyme has been implicated in multiple cancers due to its involvement in DNA replication, transcription, and chromatin remodeling [30,42,43]. Etoposide and the anthracycline chemotherapeutic drugs—doxorubicin, daunorubicin, and epirubicin—inhibit *TOP2A* by obstructing its ability to repair DNA strands after being cleaved [44,45,46]. A recent study has found that high rates of HER2 amplification are observed in PDAC patients with *TOP2A* amplification [47]. More recently, Enhertu (fam-trastuzumab deruxtecan-nxki) has become available for the treatment of unresectable or metastatic HER2-positive breast cancer, and this drug attacks both HER2 and topoisomerase in cancer cells [48,49]. In future studies, it will be necessary to investigate the efficacy of these anticancer drug combinations in patients with PDAC highly expressing *TOP2A* and/or HER2.

A previous report indicates that several transcription factors closely control the expression of *TOP2A* [30]. We further investigated the clinical significance of *TOP2A* transcriptional regulators—*HDAC1, HDAC2, HMGB1, HMGB2, NFYA, NFYB, NFYC*, and *SP1*—in PDAC clinical specimens. The genes (except for *HDAC2*) were upregulated in PDAC tissues, and the elevated expression of two genes (*HMGB2* and *SP1*) significantly predicted the short-term survival of the patients. High-mobility group proteins (HMGs) are the second most abundant chromatin proteins, and HMG-family proteins contribute to transcriptional fine-tuning in response to rapid environmental changes [50]. Upregulations of high mobility group B (HMGB) members are associated with malignant phenotypes (e.g., uncontrolled cell proliferation, avoidance of apoptosis, and metastasis) in several types of cancers [50]. In PDAC cells, silencing *HMGB2* has shown the reduction of the hypoxia-inducible factor 1α (HIF1α) protein and inhibits the HIF1α-mediated glycolytic process [50]. Specific protein 1 (*SP1*) is a zinc-finger transcription factor that binds to GC-rich motifs of many promoters. Previous studies have shown that the high expression of *SP1* is correlated with aggressive behavior in many types of cancers and decreased survival rate of the patients [51]. In PDAC cells, *SP1* transcriptionally activates COX-2 expression, and this event promotes the angiogenesis of PDAC [51]. A recent study has shown that *SP1* induces the expression of lysyl oxidase-like 2 (LOXL2) and activates the EMT process. As a result, the aberrant expression of *SP1* is involved in the acquisition of the invasion and migration abilities of PDAC cells [52]. We confirmed the expression of *HMGB2* and *SP1* in PDAC clinical specimens by immunohistochemical staining. Our results indicated that the overexpression of several transcriptional regulators coordinately enhanced the expression of *TOP2A* in PDAC cells. Therefore, the overexpression of these transcription factors might contribute to the malignant transformation of PDAC and might serve as an index of drug resistance.

## 4. Materials and Methods

### 4.1. Collection of Human Clinical PDAC Specimens, Pancreatic Tissue Specimens, and PDAC Cell Lines

The present study was approved by the Bioethics Committee of Kagoshima University (Kagoshima, Japan; approval no. 160038 28-65, 4 September 2016). Written prior informed consent and approval were obtained from all patients.

In this study, 27 clinical samples were collected from PDAC patients who underwent resection at Kagoshima University Hospital from 1997 to 2016. Sixteen normal pancreatic tissue specimens were collected from noncancerous regions. The clinical samples were staged according to the American Joint Committee on Cancer/Union Internationale Contre le Cancer (UICC) TNM classification. The total RNA of clinical samples were extracted from frozen specimens. Immunohistochemistry about PDAC was prepared from formalin-fixed and paraffin-embedded. The clinical features of the PDAC specimens are shown in Appendix A.

We used two PDAC cell lines: SW1990, purchased from the American Type Culture Collection (Manassas, VA, USA), and PANC-1, purchased from RIKEN Cell Bank (Tsukuba, Ibaraki, Japan).

### 4.2. RNA Extraction and Quantitative Real-Time Reverse Transcription Polymerase Chain Reaction (qRT-PCR)

The methods for RNA extraction from clinical specimens and cell lines and for qRT-PCR have been described previously [32,33,41]. The TaqMan probes and primers used in this study are listed in Appendix A.

### 4.3. Transfection of miRNAs, siRNAs, and Plasmid Vectors into PDAC Cells

The transfection procedures of miRNAs, siRNAs, and plasmid vectors into PDAC cells by using Opti-MEM and Lipofectamine RNAiMAX transfection regent have been described previously [32,33,41]. The reagents used in this study are listed in Appendix A.

### 4.4. Incorporation of miRNAs (miR-30c-5p and miR-30c-2-3p) into RNA-Induced Silencing Complex (RISC) by Agonaute-2 (AGO2) Immunoprecipitation

For the measurement of incorporated miRNAs into RISC in PDAC cells, we applied Ago2 immunoprecipitation by using a microRNA Isolation Kit, Human Ago2 (Wako Pure Chemical Industries, Ltd., Osaka, Japan). The number of Ago2-conjugated miRNAs were assessed by qRT-PCR assay. The experimental procedure has been described in previous studies [32,33,41].

### 4.5. Functional Assays in PDAC Cells (Cell Proliferation, Migration, and Invasion Assays)

The procedures for the functional assays in cancer cells (proliferation, migration, and invasion) are described in our previous publications [32,33,41]. PANC-1 and SW1990 cells were transfected with 10 nM miRNA or siRNA. 

For cell proliferation assays, cells were transfected in 96-well plates. PANC-1 cells were plated at 4 × 10^3^ cells per well, and SW1990 cells were plated at 6 × 10^3^ cells per well. After 72 h, cell proliferation was evaluated using XTT assays. 

Migration assays were performed with uncoated transwell polycarbonate membrane filters, and the invasion assays were performed in modified Boyden chambers without any other proteins about the extracellular matrix. For migration and invasion assays, PANC-1 cells at 2 × 10^5^ and SW1990 cells at 3 × 10^5^ were transfected in 6-well plates. After 72 h, PANC-1 and SW1990 cells were adjusted to 2.5 × 10^5^ and were added into each chamber. After 48 h, the cells of the lower surface were counted for analysis. All experiments were performed in triplicate.

### 4.6. Identification of miR-30c-5p and miR-30c-2-3p Targets in PDAC Cells

We selected putative target genes having binding sites for *miR-30c-5p* and *miR-30c-2-3p* using TargetScanHuman ver.7.2 (http://www.targetscan.org/vert_72/; data was downloaded on 13 July 2018). Our microarray data (from *miR-30c-5p* or *miR-30c-2-3p* transfected cells) were deposited in the GEO repository under accession number GSE106791. To determine the genes upregulated in the PDAC clinical specimens, we obtained expression data from the Gene Expression Omnibus (GEO) database (GSE15471). The expression data of 39 pairs of specimens are stored in this dataset.

### 4.7. Clinical Database Analysis of miRNA Target Genes in PDAC Clinical Specimens

For the analysis of gene expression between normal and cancerous tissues, we utilized the GSE15471 datasets obtained from the GEO database. In brief, GSE15471 contained mRNA array data from 36 PDAC tumors and matching normal pancreatic tissue samples, which were obtained using Affymetrix U133 Plus 2.0 whole-genome chips. The expression levels were shown as signal intensity, and for each gene with multiple probes, the mean value was used.

For the Kaplan–Meier survival analysis, we downloaded TCGA clinical data (TCGA, Firehose Legacy) from cBioportal (https://www.cbioportal.org). Gene expression grouping data for each gene were collected from OncoLnc (http://www.oncolnc.org). R version 4.0.2 was used for statistical analyses.

### 4.8. Plasmid Construction and Dual-Luciferase Reporter Assays

Plasmid vectors containing wild-type sequences of *miR-30c-2-3p* binding sites in the 3′-UTR of *TOP2A* or deletion sequences of *miR-30c-2-3p* binding sites in the 3′-UTR of *TOP2A* were prepared. The detailed procedures for transfection and dual-luciferase reporter assays have been described in our previous studies [32,33,41]. The reagents used in this study are listed in Appendix A.

### 4.9. Western Blotting and Immunohistochemistry

The procedures for Western blotting and immunohistochemistry have been described in our previous publications [32,33,41]. The antibodies used in the present study are listed in Appendix A.

### 4.10. Statistical Analyses

Mann-Whitney U tests were applied for comparisons between two groups. For multiple groups, one-way analysis of variance and Dunnett’s test were applied. These analyses were performed with JMP Pro 14 (SAS Institute Inc., Cary, NC, USA).

## 5. Conclusions

We demonstrated that both strands of pre-30c (miR-30c-3p and *miR-30c-2-3p*) acted as tumor-suppressive miRNAs in PDAC cells. A total of 10 genes (*YWHAZ*, *F3*, *TMOD3*, *NFE2L3*, *ENDOD1*, *ITGA3*, *RRAS*, *PRSS23*, *TOP2A*, and *LRRFIP1*) were found to be regulated by these miRNAs, and high expression levels were significantly predictive of short survival times in patients with PDAC. We further investigated the expression control of *TOP2A* in PDAC cells. The overexpression of several transcriptional factors and downregulation of *miR-30c-2-3p* contributed to the upregulation of *TOP2A* in PDAC cells. Our miRNA-based strategy thus provides novel insight, contributing to our overall understanding of the molecular pathogenesis of PDAC.

## Figures and Tables

**Figure 1 cancers-12-02731-f001:**
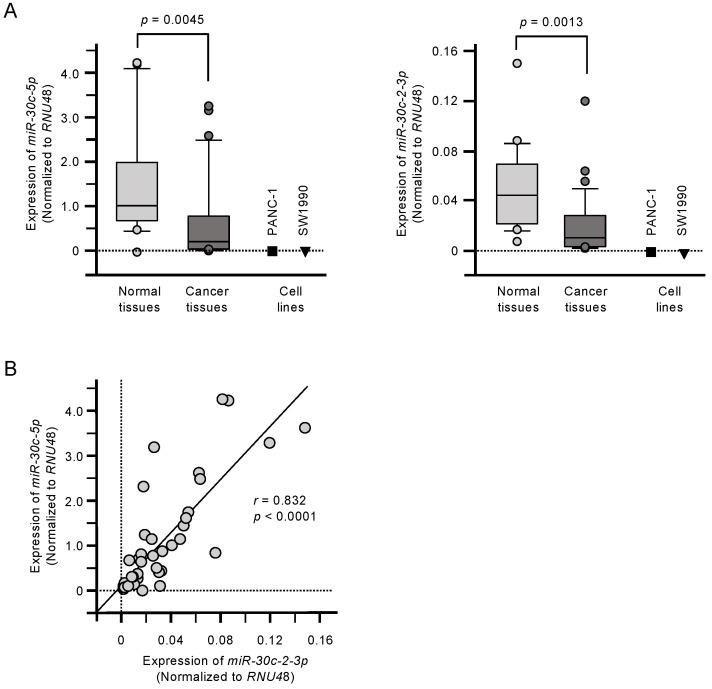
Downregulation of *miR-30c-5p* and *miR-30c-2-3p* in pancreatic ductal adenocarcinoma (PDAC). (**A**) Expression levels of *miR-30c-5p* and *miR-30c-2-3p* in PDAC clinical specimens and cell lines (PANC-1 and SW1990). Data were normalized to the expression of RNU48. (**B**) Spearman’s rank tests showed positive correlations between the expression levels of *miR-30c-5p* and *miR-30c-2-3p* in clinical specimens.

**Figure 2 cancers-12-02731-f002:**
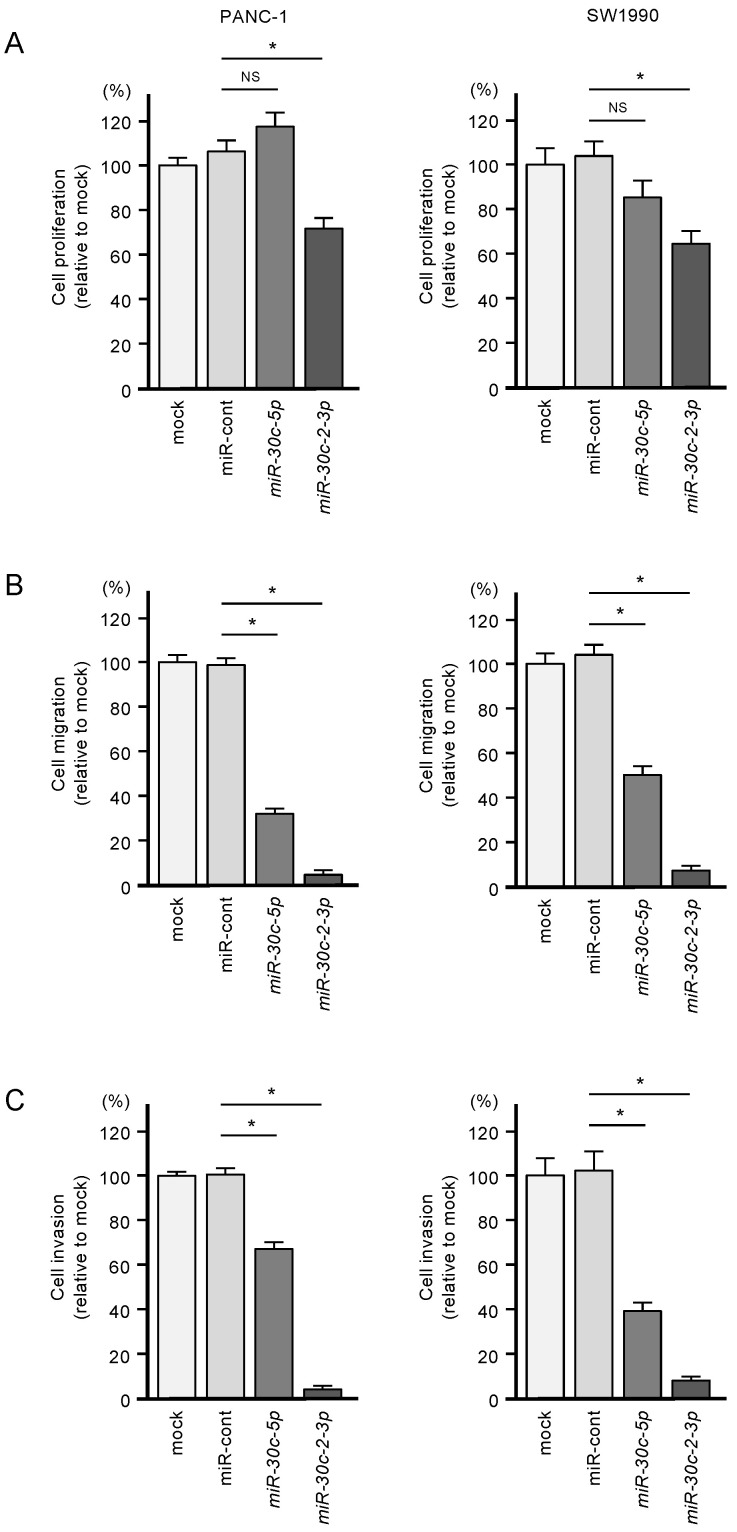
Tumor-suppressive roles of *miR-30c-5p* and *miR-30c-2-3p* in PDAC cell lines. (**A**) Cell proliferation was measured by the XTT assay. Data were collected 72 h after miRNA transfection (* *p* < 0.0001). (**B**) Cell migration was assessed with a membrane culture system. Data were collected 48 h after seeding the cells into chambers (* *p* < 0.0001). (**C**) Cell invasion was determined 48 h after seeding miRNA-transfected cells into chambers using Matrigel invasion assays (* *p* < 0.0001).

**Figure 3 cancers-12-02731-f003:**
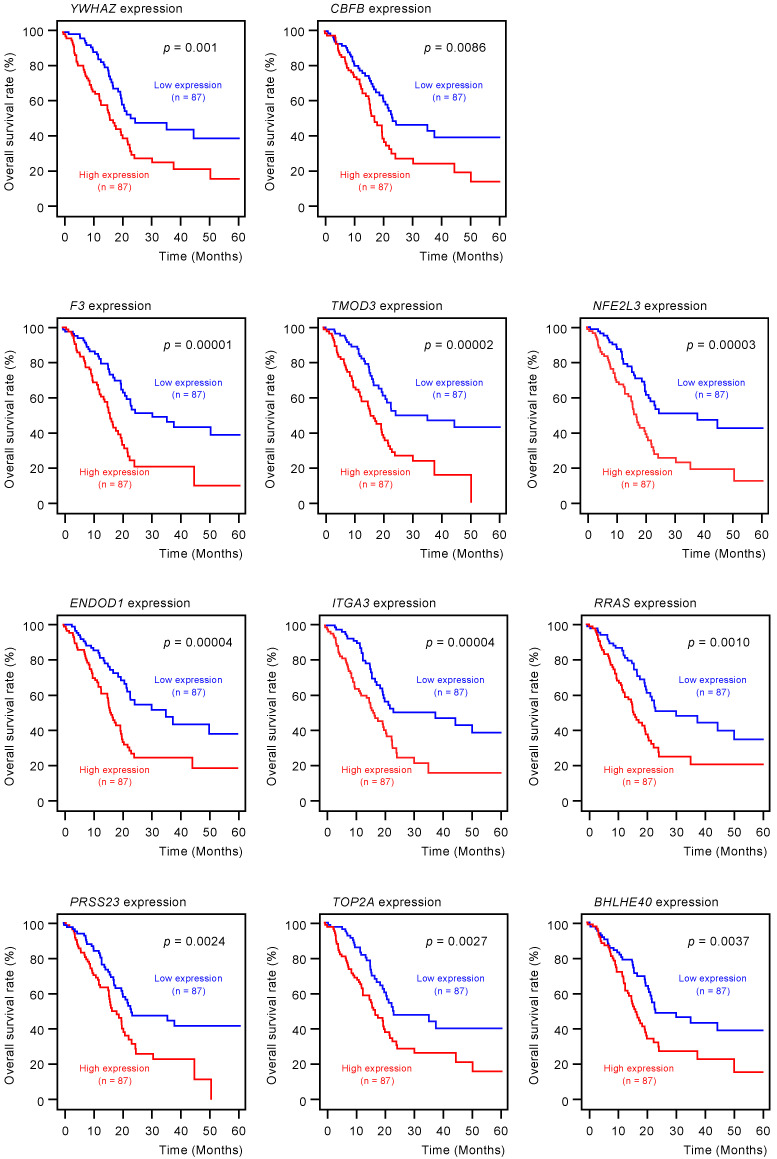
Clinical significance of *miR-30c-5p* and *miR-30c-2-3p* target genes by The Cancer Genome Atlas (TCGA) database analysis. Among the putative target genes of *miR-30c-5p* and *miR-30c-2-3p* regulation in PDAC cells, high expression levels of 18 genes (*YWHAZ*, *CBFB*, *F3*, *TMOD3*, *NFE2L3*, *ENDOD1*, *ITGA3*, *RRAS*, *PRSS23*, *TOP2A*, *BHLHE40, CHST11, IL1RAP, DIEXF, CORO1C, R**TP4, AP1S3*, and *LRRFIP1*) were found to significantly predict worse prognoses in patients with PDAC (*p* < 0.01). Kaplan–Meier curves of the 5-year overall survival rate for each gene are presented.

**Figure 4 cancers-12-02731-f004:**
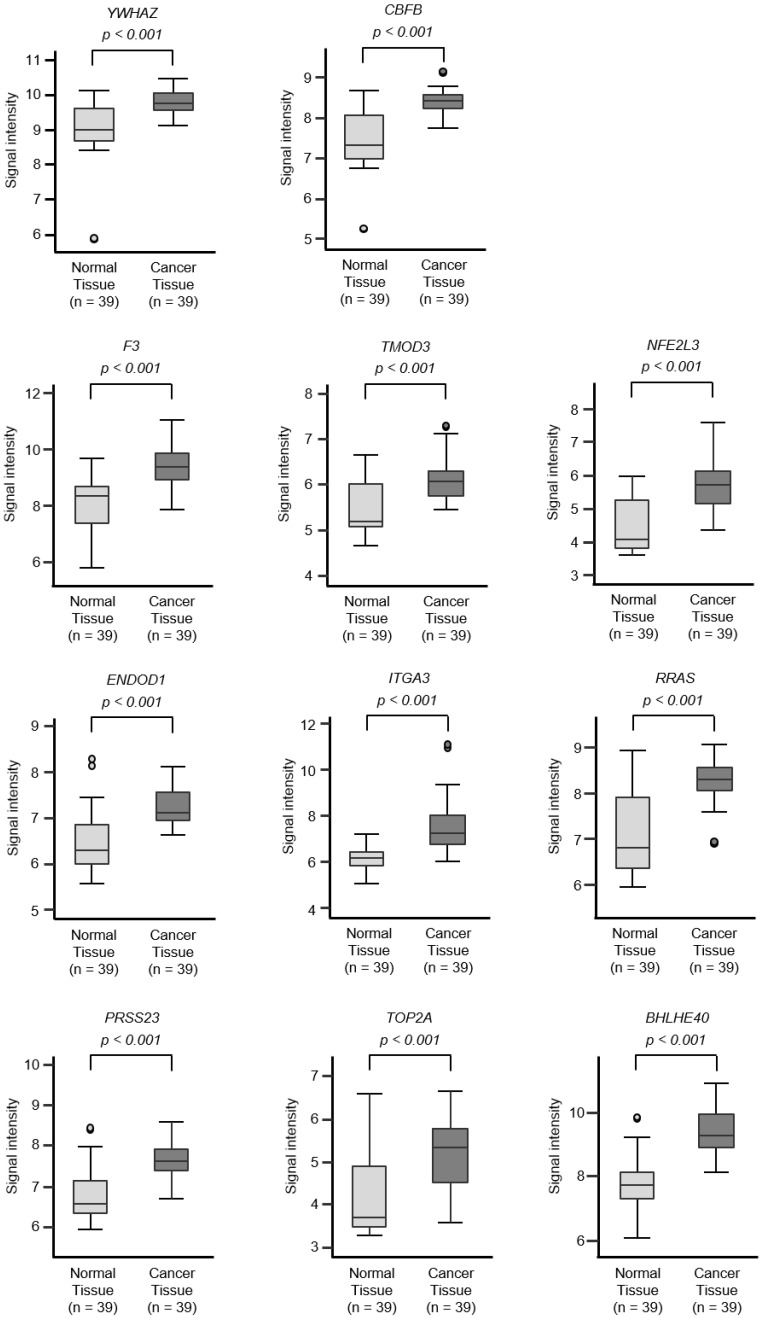
Expression levels of 18 target genes (that predicted 5-year survival) by *miR-30c* regulation in PDAC clinical specimens by TCGA analyses. Expression levels of 18 target genes of *miR-30c* (Figure 3) were evaluated using TCGA database analyses. All genes were found to be upregulated in PDAC tissues (*n* = 39) compared to normal tissues (*n* = 39).

**Figure 5 cancers-12-02731-f005:**
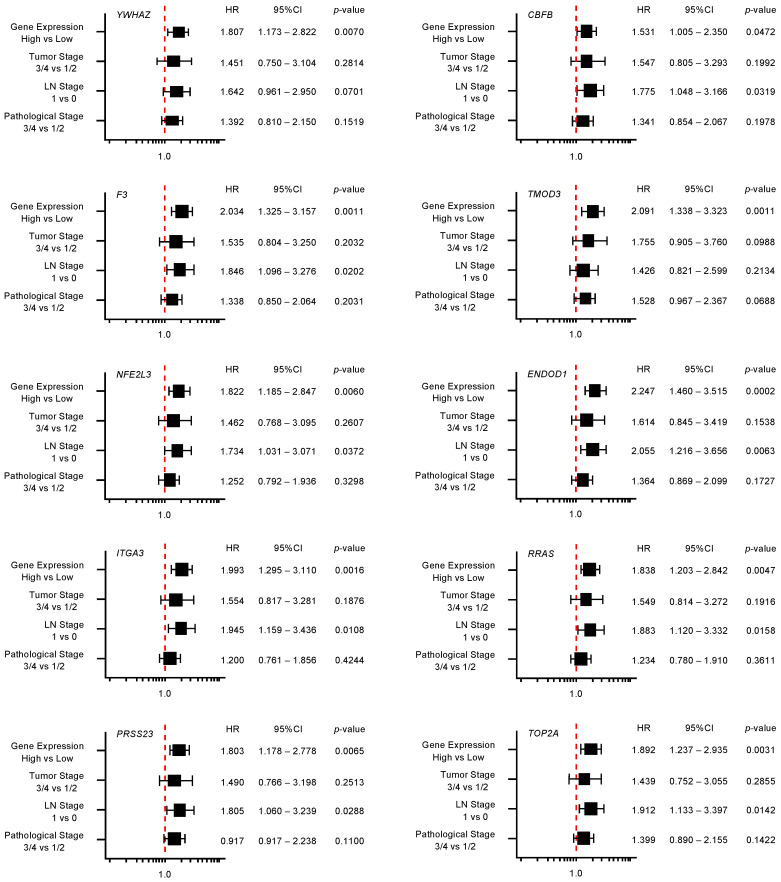
Forest plot of multivariate analysis of 18 target genes (that predicted 5-year survival) of miR-30c. The multivariate analysis determined that the expression levels of 10 genes (*YWHAZ*, *F3*, *TMOD3*, *NFE2L3*, *ENDOD1*, *ITGA3*, *RRAS*, *PRSS23*, *TOP2A*, and *LRRFIP1*) were independent prognostic factors for 5-year overall survival after the adjustment for tumor stage, lymph node metastasis, and pathological stage (*p* < 0.01).

**Figure 6 cancers-12-02731-f006:**
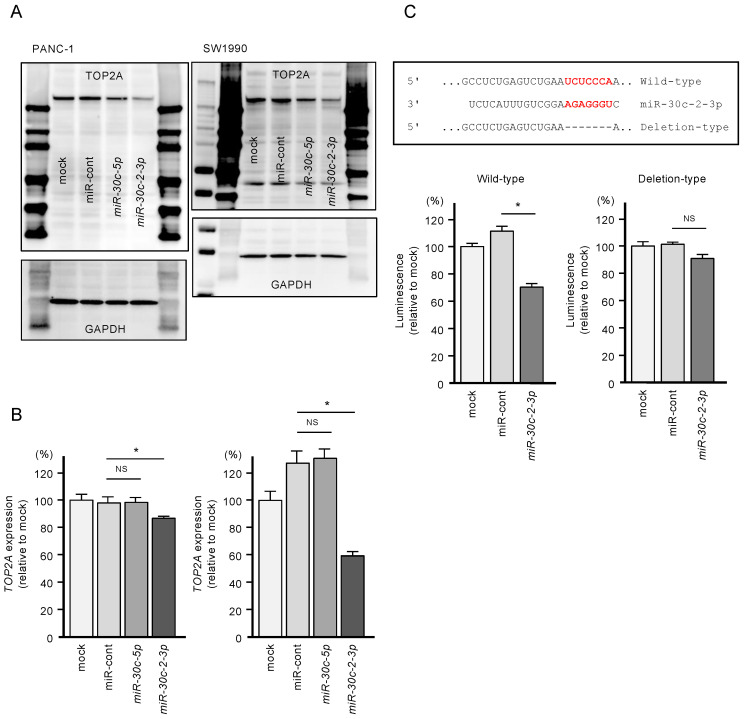
Direct regulation of *TOP2A* by *miR-30c-2-3p* in PDAC cells. (**A**) Protein expression levels of *TOP2A* were significantly reduced by *miR-30c-2-3p* transfection into PANC-1 and SW1990 cells (72 h after transfection). Images of whole Western blot gels are shown. GAPDH was used as an internal control. (**B**) mRNA expression levels of *TOP2A* were significantly reduced by *miR-30c-2-3p* transfection into PANC-1 and SW1990 cells (* *p* < 0.0001). (**C**) TargetScan database analysis showed that one putative binding site of *miR-30c-2-3p* was annotated in the 3′-UTR region of *TOP2A*. Dual-luciferase reporter assays showed that luminescence activity was reduced by co-transfection with wild-type vector (containing *miR-30c-2-3p* binding sites) and *miR-30c-2-3p* in PANC-1 cells. Normalized data were calculated as the ratios of Renilla/firefly luciferase activity (* *p* < 0.0001).

**Figure 7 cancers-12-02731-f007:**
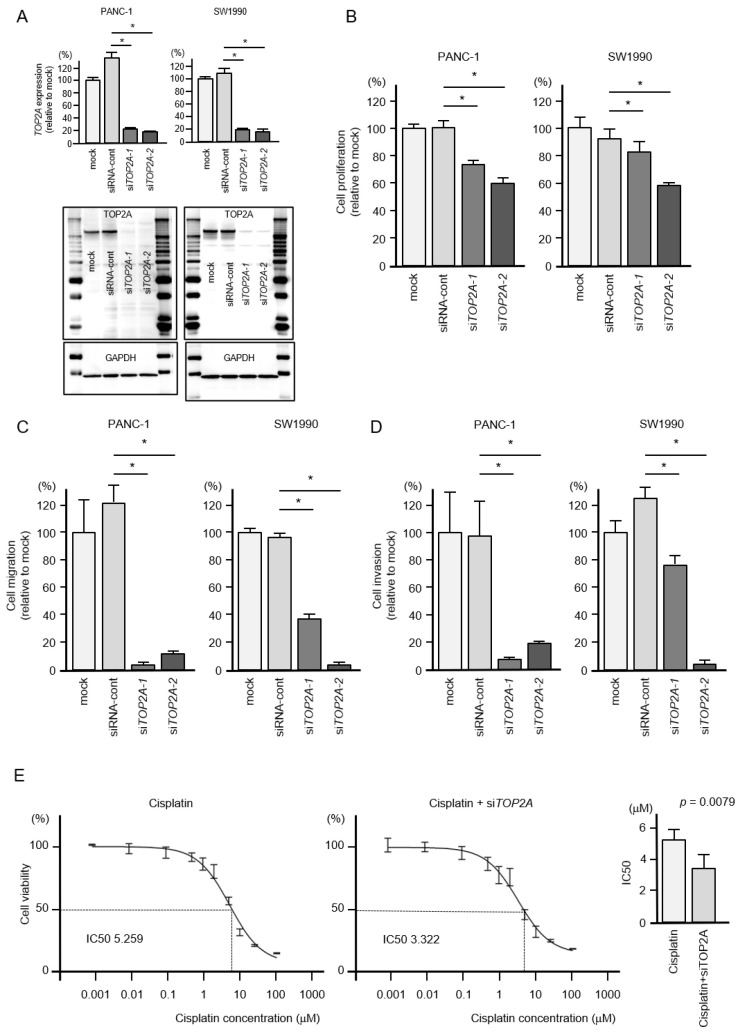
Effects of *TOP2A* knockdown in PDAC cells. (**A**) Validation of *TOP2A* knockdown efficiency in siRNA-transfected PDAC cells. The mRNA expression levels of *TOP2A* were significantly reduced by *miR-30c-2-3p* transfection into PANC-1 and SW1990 cells. The protein expression levels of *TOP2A* were significantly reduced by transfection of two siRNAs (si*TOP2A*-1 and si*TOP2A*-2) into PANC-1 and SW1990 cells (measured 72 h after transfection). Images of whole Western blot gels are shown. GAPDH was used as an internal control. (**B**) Cell proliferation was measured by the XTT assay. Data were collected 72 h after siRNA transfection (* *p* < 0.0001). (**C**) Cell migration was measured using a membrane culture system. Data were collected 48 h after seeding the cells into chambers (* *p* < 0.0001). (**D**) Cell invasion was determined 48 h after seeding miRNA-transfected cells into chambers using Matrigel invasion assays (* *p* < 0.0001). (**E**) In PDAC cells, si*TOP2A* increased the sensitivity to the anticancer drug cisplatin. Half maximal inhibitory concentration (IC_50_) values were worked out by using GraphPad Prism8 software.

**Figure 8 cancers-12-02731-f008:**
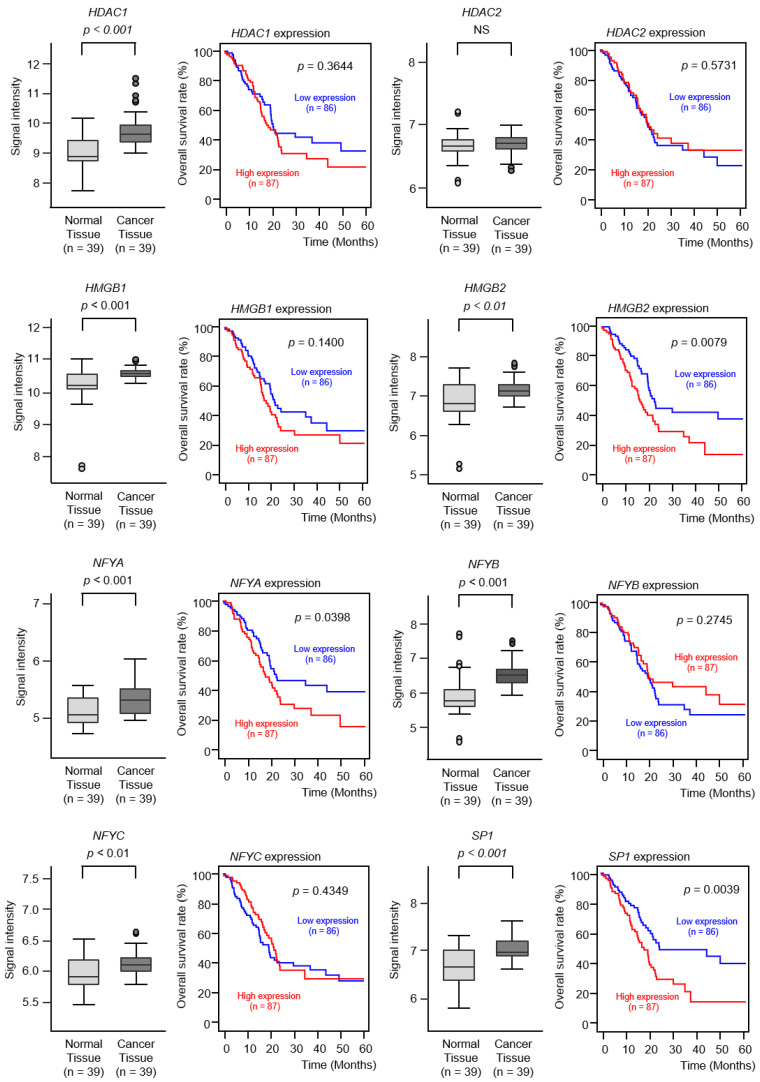
Expression levels and clinical significance of *TOP2A* transcriptional regulators by TCGA analyses. The expression levels of 8 genes (*HDAC1, HDAC2, HMGB1, HMGB2, NFYA, NFYB, NFYC*, and *SP1*) involved in *TOP2A* transcriptional regulation were evaluated by TCGA database analyses. All genes (except for *HDAC2*) were upregulated in PDAC tissues (*n* = 39) compared to normal tissues (*n* = 39). The overall survival rates were evaluated using the TCGA database. Among the 8 genes identified (*HDAC1, HDAC2, HMGB1, HMGB2, NFYA, NFYB, NFYC*, and *SP1*), high expression levels of 2 genes (*HMGB2* and *SP1*) were found to be significantly predictive of worse prognoses in patients with PDAC (*p* < 0.01). Kaplan–Meier curves of the 5-year overall survival rate for each gene are presented.

**Figure 9 cancers-12-02731-f009:**
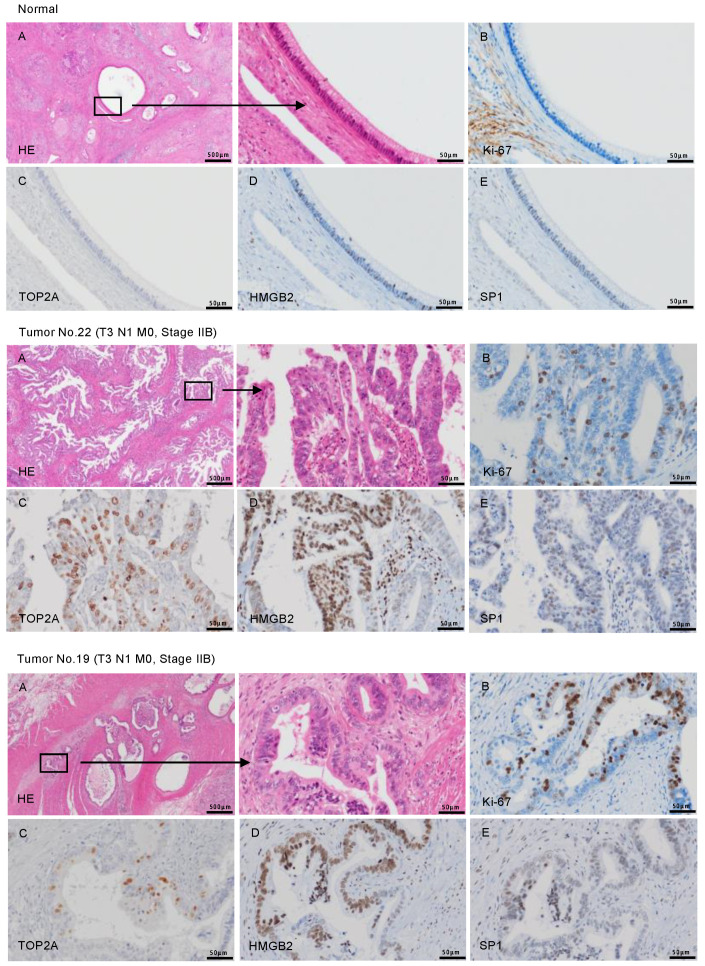
The overexpression of *TOP2A*, *HMGB2*, and *SP1* in PDAC clinical specimens determined by immunohistochemical staining. Representative immunohistochemical images for *TOP2A*, *HMGB2*, and *SP1* in clinical samples are shown. (**A**) H&E stained sections of human PDAC tissue. Areas in the boxed region at left are shown magnified at right. (**B**) Ki-67 was used as a cell proliferation marker. (**C**) Immunohistochemical staining for *TOP2A* showed patchy nuclear signals in PDAC. (**D**,**E**) Immunohistochemical staining for *HMGB2* (**D**) and *SP1* (**E**) was found in the nuclei of cancer cells. The patient numbers are shown in Appendix A.

**Table 1 cancers-12-02731-t001:** Putative target genes by *miR-30c-5p* and *miR-30c-2-3p* regulation in PDAC cells.

**Putative Target Genes by *miR-30c-5p* Regulation in PDAC Cells**							
**Entrez** **Gene**	**Gene** **Symbol**	**Gene Name**	**GEO ^1^** **FC ^2^**	**GEO** **log_2_ FC**	**PANC-1** ***miR-30c-5p*** **Transfectant** **FC**	**PANC-1** ***miR-30c-5p*** **Transfectant** **log_2_ FC**	**Total** **Binding** **Sites**
7534	*YWHAZ*	tyrosine 3-monooxygenase/tryptophan 5-monooxygenase activation protein, zeta polypeptide	2.11	1.08	−1.80	−0.85	1
865	*CBFB*	core-binding factor, beta subunit	2.23	1.16	−2.33	−1.22	2
**Putative Target Genes by *miR-30c-2-3p* Regulation in PDAC Cells**							
**Entrez** **Gene**	**Gene** **Symbol**	**Gene Name**	**GEO** **FC**	**GEO** **log_2_ FC**	**PANC-1** ***miR-30c-2-3p*** **Transfectant** **FC**	**PANC-1** ***miR-30c-2-3p*** **Transfectant** **log_2_ FC**	**Total** **Binding** **Sites**
2152	*F3*	coagulation factor III (thromboplastin, tissue factor)	2.63	1.39	−1.62	−0.69	1
29766	*TMOD3*	tropomodulin 3 (ubiquitous)	2.05	1.03	−2.80	−1.48	2
9603	*NFE2L3*	nuclear factor, erythroid 2-like 3	2.85	1.51	−2.13	−1.09	3
23052	*ENDOD1*	endonucleasedomain containing 1	2.26	1.18	−1.56	−0.64	1
3675	*ITGA3*	integrin, alpha 3 (antigen CD49C, alpha 3 subunit of VLA-3 receptor)	2.51	1.33	−2.56	−1.36	2
6237	*RRAS*	related RAS viral (r-ras) oncogene homolog	2.21	1.15	−1.89	−0.92	1
11098	*PRSS23*	protease, serine, 23	2.75	1.46	−1.92	−0.94	1
7153	*TOP2A*	topoisomerase (DNA) II alpha 170kDa	2.89	1.53	−2.28	−1.19	1
8553	*BHLHE40*	basic helix-loop-helix family, member e40	3.22	1.69	−1.56	−0.64	1
50515	*CHST11*	carbohydrate (chondroitin 4) sulfotransferase 11	2.95	1.56	−1.80	−0.85	1
3556	*IL1RAP*	interleukin 1 receptor accessory protein	2.01	1.01	−1.80	−0.83	1
27042	*DIEXF*	digestive organ expansion factor homolog (zebrafish)	2.11	1.08	−1.62	−0.70	5
23603	*CORO1C*	coronin, actin binding protein, 1C	2.96	1.57	−2.90	−1.54	1
64108	*RTP4*	receptor (chemosensory) transporter protein 4	2.27	1.18	−5.89	−2.56	1
130340	*AP1S3*	adaptor-related protein complex 1, sigma 3 subunit	2.09	1.06	−4.18	−2.05	3
9208	*LRRFIP1*	leucine rich repeat (in FLII) interacting protein 1	2.65	1.40	−2.43	−1.28	2

^1^ Gene Expression Omnibus, ^2^ Fold change.

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
