# Peer review of "Molecular Pathogenesis of Pancreatic Ductal Adenocarcinoma: Impact of *miR-30c-5p* and *miR-30c-2-3p* Regulation on Oncogenic Genes"

_cancers, 2020, doi:10.3390/cancers12102731_

Round 1

Reviewer 1 Report

Tanako and co-workers describe their findings clearly and the conclusions are properly based on the experimental results or the meta-Analysis of Patient data.

To me, the results sound very intriguing, that both guide and passenger strand of  miR-30c serves as Tumor suppressor.

I have only some minor Questions or concerns:

(1) the authors should give more Detail on the Migration and Invasion assays. Which ECM protein did they use to coat the polycarbonate filter? This may Change the Outcome of the Migration Assay. Did they use an integrin a3b1 integrin Ligand? Moreover, the description of the Invasion Assay (= modified Boyden chamber) is very slim. The authors should give more Details.

(2) In Figures 4 and  9, the authors tested the Expression of several genes in 39 samples. In the method section, they Mention 31 and 36 Patient samples. How do These numbers go together?

(3) in Figure 8B, the Inhibition curves are very close. Did the authors compared the two curves for significance of difference? Also, the IC50 values should be mentioned with the Standard Deviation, to get the first Impression About a potential difference?

(4) also in Figure 8: the authors may want to expand the Abbreviation CDDP to Cisplatin in the figure to facilitate a fast understanding.

(5) minor: line 342, the word "of" is missing between 'data' and 'Gene'.

Reviewer 2 Report

Tanaka et al. provide a study about the role of miRNA in PDAC.

In general, the manuscript is well-written and interesting new results are provided. Nevertheless, we have some topics that should be addressed before acceptance for publication.

  1. In Figure 1 the sample size (n) should be mentioned.
  2. In Figure 2 and 8, the number how often the in vitro experiment was repeated.
  3. We request the authors to place the Table 1A, 1B in supplementary section.
  4. Instead of presenting the 19 putative target genes of the miR-30c-5p and miR-30c-2-3p, it would be interesting to categorize them based on functionalities (e.g., proliferation migration, invasion or being part of a signaling pathway).
  5. In order to validate the TCGA database analysis shown in Figure 3, the authors should evaluate the expression of the 19 genes both at mRNA and protein levels in the clinical specimens. They should also include the two cell lines they have.
  6. Since PDAC is more prevalent in elderly, we suggest the authors to add ‘age’ (young vs. old) as a factor while performing the multivariate analysis.
  7. In Figure 3 legend, the authors mentioned 19 genes that show correlation between expression and survival rate, but in the figure, only 18 are shown. Please address the discrepancy.
  8. We request the author to defend in detail on why they choose TOP2A based on already known reports. Else the further analysis based on TOP2A lacks proper rationale and continuity.
  9. The validation of TOP2A knockdown efficiency in siRNA-transfected PDAC cells shown in Figure 7 should go in supplementary section since the knockdown is itself not a functional readout (assays done with siRNA-transfected cells), rather just a step/method to the readouts.
  10. For Figure 10, we ask the authors to present on the HMGB2 expression and SP1 expression as these two are significantly predictive of the 5-year survival rate of patients with PDAC. The rest of the gene expression data should be place in supplement
  11. For Figure 11,
    1. The authors are required to present stained section from normal tissue as well.
    2. Instead of presenting the histology and immunohistochemistry in 3 panels, we request to show one representative panels with high resolution images. If 3 panels are shown – what is the difference between these 3 cases? If none, the two additional panels could go into the suppl. Data.
    3. In addition, we request to add a scale and make sure all images are at the same magnification.
    4. And most importantly, the expressions of TOP2A, HMGB2, and SP1 protein (from cancer and normal tissues) should be evaluated semi-quantitatively on all specimen and presented as a plot. Such an analysis can be performed by an experienced pathologist or also by using a computer assisted analysis.
  12. It would be interesting, if there are any relationsships or redundancies of the detected relevant genes with the gene signatures/subtypes detected in PDAC in the last years.

Reviewer 3 Report

The manuscript is formally elaborated precisely, the topic is interesting. A bit surprising is the complete formal agreement with works published in the past.

It would be interesting to see miR-30c-5p and miR-30c-3p expression after transfection as only data on proliferation, migration, and invasion are shown.

Tables 1A and 1B are too extensive to be published in the article main body, it would be more appropriate to list them as supplementary material. Rather than a list of genes, the authors should show a microRNA-mRNA interaction map/network.

I wonder if 19 target genes were also up-regulated in the tumor tissues of the patients included in the study. It would be beneficial if the extensive in silico data were verified on PDAC tumor tissues.   

Why did the authors choose TOP2A only for validation when there were several different genes significantly associated with prognosis?  What was the main reason for this decision.

Evaluation of TOP2A, HMGB2, and SP expression was done by immunochemistry, but only representative immunohistochemical images were shown. Was expression homogeneous in all PDAC tumors? Was it compared to healthy tissues?

At least a brief description of the methods would be beneficial, instead of references to published works that are linked to other references.

Round 2

Reviewer 2 Report

We thank the authors for addressing our concerns.

All our issues have been commented or changed, but there still seems to be an error with figure 9. It would be nicer to also have the Ki67 staining of normal tissue provided. And we request the authors to check the figure description - it should now also mention normal tissue and B/C seem to be switched. The D/E description should be adapted also.
